# Anticoagulant Activity of the Polysaccharide Fromgonad of Abalone *Haliotis discus hannai* Ino: The Role of Conjugate Protein

**DOI:** 10.3390/foods13244003

**Published:** 2024-12-11

**Authors:** Qinhao Liu, Siyu Yao, Siyuan Ma, Ting Zhao, Zhenyu Wang, Liming Sun, Ming Du

**Affiliations:** National Engineering Research Center of Seafood, School of Food Science and Technology, Dalian Polytechnic University, Dalian 116034, Chinawangzy0506@outlook.com (Z.W.)

**Keywords:** abalone gonad, polysaccharide, anticoagulant activity, conjugate protein

## Abstract

Few studies are concerned with the effect of the conjugat protein on the bioactivities of the abalone gonad polysaccharide (AGP). In this study, a series of treatments, including raw material (female and male) defatting, extraction temperature (25–121 °C), proteolysis, ultrafiltration, and ethanol precipitation, was conducted to investigate the role of the conjugate protein on AGP anticoagulant activity. All AGP extracts significantly prolonged activated partial thromboplastin time (APTT) and thrombin time (TT). The strongest was observed in the female AGPs prepared at 50 and 121 °C. The most active is located at 30–300 kDa by ultrafiltration. After being exposed to neutral protease, quick shortening of APTT and TT was found in all AGPs. Further ethanol precipitating of found the longest APTT in the sediment, which contains most polysaccharides and proteins. Defatting lowered the activity of female AGP but increased that of males. Proteolysis also significantly weakened the clotting factor inhibition effect of the 50 °C female AGP, but heating seemed not affect the effect. Five fractions were obtained after the 50 °C female AGP was subjected to ion exchange column. Fraction V, with the highest protein and medium polysaccharide content, showed the strongest anticoagulant effect and was also much higher than AGSP, which was obtained by multi-step proteolysis. The findings supported positive effect of the conjugate protein in AGP anticoagulant activity.

## 1. Introduction

Abalone is flavorful and nutritional. Traditional Chinese medical books also documented the healing effect of abalone meat and shell. Therefore, it has been a kind of aquatic product with high commercial value. And abalone farming has been thriving for years. The abalone broodstock in China alone was 128,051,400,000 grains, and abalone production reached up to 244,991 tons in 2023 (China Fisheries Yearbook). Due to the high price of abalone, deep processing and comprehensive utilization of by-products (visceral, mainly gonad) have been the concern of farmers and factory owners.

Abalone feeds on seaweed, with polysaccharide-degrading enzymes in its digestive tract [1,2,3]. Seaweeds are rich in dietary fiber and polysaccharides, so it is logical that polysaccharides became the target for comprehensive utilization of abalone gonad. Component analysis also found that the polysaccharide content in abalone gonads is high, next to the protein (dry basis) [4]. Abalone gonad polysaccharides (AGP) contain heteropolysaccharides with sulfate groups and glyoxalate. In vitro and in vivo studies also indicated that AGPs possessed a variety of biological activities, such as antitumor [5,6], antioxidant [7,8,9], immunomodulation [10,11], anticoagulation [12,13,14,15], gastrointestinal function regulation [16,17], glucose regulation/diabetic somatostatin regulation [18], osteogenesis [19], anti-atherosclerosis and hypolipidemic [20], and antiviral [21,22] activity. Therefore, the efficient utilization of abalone gonad and developed AGP products are worthy of further study and can provide experience and methods for the efficient utilization of other shellfish by-products.

Polysaccharide is always conjugate with protein, and the extracted polysaccharide would contain much protein. Therefore, most studies take it for granted to remove the conjugate protein as much as possible to increase the purity of the extracted polysaccharide. And protease has long been used singly or combined to remove the protein. The solubility of polysaccharides in ethanol is very low. So, proteolysis combined with ethanol precipitation has been found to be effective in removing protein and enriching polysaccharides. However, large quantities of ethanol pose a great cost and safety challenge to polysaccharide production. In addition, other methods were also used or attempted to remove protein. Sevag’s reagent (n-butanol and chloroform, 1:4 v/v) is considered effective in removing free protein from the proteolysate originating from proteolysis [23]. However, due to the toxicity, the polysaccharides prepared by Sevag’s reagent can only be used for basic studies, such as structural analysis. Deep eutectic solvent (Choline chloride combined with ethylene glycol in a molar ratio 1:3) was also attempted aiming to increase the polysaccharide yield [24]. The problem with this method is that the water content in the system needs to be controlled below 25%, and any higher water content will reduce the extraction rate. A method of attapulgite-based silk fibroin composite aerogel was also attempted to remove protein from polysaccharides of *Ruditapes philippinarum* [25], and the cost and processing make it far from scale application.

Reducing the protein content and increasing the polysaccharide content/purity is meaningful for the structural analysis of the polysaccharide backbone and structure-activity study. However, the effect of coupling proteins on polysaccharide bioactivity has not yet been elucidated. The optimal content of the coupling protein need to be investigated to obtain the maximum bioactivity of the polysaccharide. In a previous study, a high anticoagulant activity was incidentally found in a crude AGP extract obtained by the simple water extraction method without enzymatic digestion. The activity was much higher than that of the AGP obtained by multiple proteolysis. This phenomenon inspired us to revisit the necessity and effectiveness of deproteinization treatment.

In the present study, the female and male gonads of the most cultured abalone, *Haliotis discus hannai* Ino, were used as raw materials. AGP water extract was prepared at temperatures from 25 to 121 °C. The effect of deproteinization and protein-denaturing treatments, such as proteolysis, ethanol precipitation, heating, and acetone defatting, was investigated on the anticoagulant activity of AGP extract. Ion exchange column chromatography was also conducted to further study the correlation between anticoagulant activity and protein and polysaccharide content of the fractions. Other manipulations, such as ultrafiltration, were applied to optimize the production process of AGP from gonad.

## 2. Materials and Methods

### 2.1. Material Pretreatment

Frozen mixed abalone gonad was supplied by Dalian Zhangzidao Group Co. (Dalian, China). The frozen raw materials are thawed by running water. Impurities and other visceral were removed. Female and male gonads were separated, cleaned, frozen, lyophilized, powdered through a 50-mesh sieve, and stored at −30 °C for use. A portion of the lyophilized powder was defatted by acetone, with a ratio of 1:10 (m/V). The mixture was stirred at 25 °C for 3 h. After leaching, the mixture was evaporated completely in the fume hood to obtain the defatted powder of female and male gonads. Preparation plasma was in accordance with the Guidelines for the care and use of laboratory animals, as described by the National Health and Medical Research Council of China and approved by the Animal Ethics Committee of Dalian Polytechnic University (Approved protocol ID SYXK2017-0005).

### 2.2. Quantification and Characterization

#### 2.2.1. Protein and Polysaccharide Quantification

The protein content of the abalone gonad was measured by the Kjeldahl method. The bicinchoninic acid assay (BCA) method was used to determine protein content in solution samples [26]. The phenol sulfuric acid method was used for polysaccharide content determination, using glucose as the standard (y = 7.8977x + 0.0604, R^2^ = 0.9921) [27]. Sulphated polysaccharide was quantified using the 1, 9-dimethylmethylene blue method [28] using chondroitin sulfate as the standard (y = 7.8977x + 0.0604, R^2^ = 0.9921). 

#### 2.2.2. Amino Acid and Infrared Spectra Analysis

Samples of 20 mg were hydrolyzed with 6 M HCL at 110 °C for 24 h in a sealed and evacuated tube. The amino acid composition was analyzed with a Hitachi LA8080 amino-acid analyzer. Fourier transform infrared spectra of the samples were recorded using the KBr-disk method. Freeze-dried sample (10 mg) was mixed with 100 mg of dried potassium bromide and compressed to a disc (10 mm diameter). Spectra were read on FT-IR Frontier (PerkinElmer Co., Waltham, MA, USA) spectrometer in the range 4000–400 cm^−1^.

### 2.3. Abalone Gonad Polysaccharide Preparation and Processing

#### 2.3.1. Abalone Gonad Polysaccharide (AGP) Preparation

Prepare five portions of female and male abalone gonad powder, respectively, with each portion of 10 g. Dissolve the gonad powder in 100 mL of deionized water, and AGP was extracted at 25, 50, 80, 100, and 121 °C, respectively, in flasks using an oven and autoclave. AGP extract of defatted gonad was prepared only at 50 °C. After stirring for 15 min, the mixture was centrifuged for 1 h at 10,000 rpm and 4 °C. The supernatant (I) was collected for anticoagulant activity measure and following use.

#### 2.3.2. Ultrafiltration

The supernatant (I) from the female gonad was subjected to ultrafiltration, using a membrane with 30 and 100 kDa of molecular weight cutoff. Crude AGP was separated into three fractions, with molecular weight ranges of <30 kDa, 30–100 kDa, and >100 kDa, respectively. The anticoagulant activity of each fraction was measured.

#### 2.3.3. Proteolysis

Each portion of the supernatant (I) obtained at different temperatures was subjected to proteolysis by adding neutral protease (with the final activity of 3000 U/g (substrate protein)) and incubated at 50 °C for 0, 0.5, 1, 3, and 5 h. After heating at boiling water for 10 min to inactivate the protease, the mixture was centrifuged for 1 h at 10,000 rpm and 4 °C. The supernatant (II), named proteolysate, was collected for anticoagulant activity measurement. Two milliliters of the proteolysate (of female AGP extracted at 25, 50, and 121 °C) was further precipitated with 95% ethanol in a ratio of 1:3 (v:v). The mixture was kept at 4 °C for 12 h, and then centrifuged for 10 min at 4000 rpm. The supernatant (III) and sediment were lyophilized and resolved with water back to 2 mL. Polysaccharide and protein content as well as the anticoagulant activity were measured. Since the APTT prolongation activity of the proteolysate and the precipitated fraction was so strong, the two parts were diluted 5 and 10 times, respectively, while the supernatant was not diluted.

#### 2.3.4. Purification

For the AGP extracted at 50 °C, ion exchange chromatography was performed on the GE ACTA Avant 25 system (GE Co., Boston, MA, USA) equipped with a DEAE Sepharose Fast Flow column with an ultraviolet detector. The elution buffer was a gradient NaCl aqueous (0.9–100%, m/v) with a flow rate of 2.0 mL/min. A tube of peak absorbance was collected and lyophilized. Polysaccharide and protein content and anticoagulant activity were measured for each fraction, compared with AGSP.

### 2.4. Abalone Gonad Sulfated Polysaccharide (AGSP) Preparation

To compare the anticoagulant activity with the polysaccharide reported by another study, AGSP was prepared by a series of treatments, including three discontinuous times of proteolysis, enrichment, and separation by resin, two times of precipitation by ethanol, dialyzing to remove salt, and lyophilization according to Guo et al. [29].

### 2.5. Anticoagulant and Clotting Factor Activity

Anti-coagulant activity assays were performed with an activated partial thromboplastin time (APTT) reagent kit (ellagic and bovine phospholipid), CaCl_2_ solution (25 mM), prothrombin time (PT) reagent kit, and thrombin time (TT) reagent kit, which were purchased from Stago Co. (Paris, France). APTT, PT, and TT were recorded by an automated coagulometer (Stago-R, Pairs, France). In each assay, the sample was mixed with anticoagulative plasma in a volume ratio of 1:9. The activity of clotting factors VIII, IX, and XI was measured by corresponding diagnostic kit, and recorded by MC 4 plus coagulation analyzer (ABW Medizin und Technik GmbH, Lemgo, Germany). Normal saline (0.9%, m/v) was used in all the above assays.

### 2.6. Statistical Analysis

All experiments were performed in triplicate. Mean values were obtained from three individual replicates. One-way analysis of variance (ANOVA) was performed using SPSS 23.0 statistical software (SPSS Inc., Chicago, IL, USA). Data are presented as mean ± standard deviation (SD). Comparisons that yielded *p* < 0.05 were considered significant.

## 3. Results

### 3.1. Polysaccharide, Protein Content and Amino Acid Composition of Abalone Gonad

Figure 1 shows the crude protein content was 41.7% and 40.69% in female and male gonads, respectively. The polysaccharide content was 20.1% and 17.3%, respectively. The protein content is almost 2 folds of the polysaccharide, so the effect of protein port on the bioactivity of polysaccharide could not be ignored.

Table 1 shows the amino acid composition of the abalone gonad. Both female and male gonads contained seven essential amino acids (tryptophan was hydrolyzed and destroyed during sample processing, and could not be measured). The essential amino acids and flavor amino acids accounted for more than 70% of the total amino acids.

### 3.2. Effect of Extraction Temperature on Anticoagulant Activity

APTT, PT, and TT are the most commonly used items to evaluate coagulation function. APTT reflects the number and function of coagulation factors involved in the endogenous pathway and common pathway. PT reflects the exogenous pathway and common pathway. The common part of APTT and PT was TT, which reflects the level and function of thrombin and fibrinogen [30].

The effect of extraction temperature on the anticoagulant activity of AGP extract is shown in Figure 2. Compared with the control, both female and male extracts significantly prolonged APTT and TT, indicating that the extracts might inhibit some coagulation factor involved in the endogenous coagulation pathway and common pathway, or the conversion of fibrinogen to fibrin [31]. However, most extracts caused shortening of PT (except 80 °C extract), suggesting the presence of pro-coagulant components or tissue factor-like compounds in the extract, which has not been reported before. The anticoagulant activity of the female extract was generally higher than that of the male extract. The 50 °C extracts showed relatively high APTT prolongation activity. Elevated extracting temperature showed more impact on APTT, less effect on TT and lest on PT.

### 3.3. Molecular Weight Range of the Anticoagulant-Active Component

In order to confirm the components exerting anticoagulant activity are macromolecular compounds, and further locate the molecular weight (MW) distribution of the active components, the AGP extracts at different temperatures were subjected to ultrafiltration (Figure 3). The results showed that fractions at the MW range of 30–100 kDa demonstrated remarkable ATPP and TT prolonging activity. The fraction with MW range of >100 kDa and <30 kDa showed an obvious trend of shortened clotting time. Thus, the anticoagulant fractions mainly centered at 30–100 kDa. The coagulation accelerator at the part of MW > 100 kDa and <30 kDa should be removed by ultrafiltration for product developmen.

### 3.4. Effect of Proteolysis on Anticoagulant Activity

Figure 2 and Figure 3 proved that temperature and molecular weight significantly affect the anticoagulant activity of AGP extracts. Protein is known to be more sensitive to heating than polysaccharides. It is suggested that the protein of AGP is likely to have a role in the anticoagulant activity of AGP. In order to prove this speculation, neutral protease was added to the AGP extracts, and anticoagulant activity during proteolysis was examined. Figure 4 shows the APTT values of the AGP extracts from the females and the males were 176.47 s and 90 s, respectively. After adding protease, even though the mixture was immediately inactivated by dipping into the boiling water bath for 10 min, the APTT value was significantly and rapidly decreased. With the proteolysis time extended, the anticoagulant activity of the proteolysate showed a small-scale fluctuation.

### 3.5. Anticoagulant Activity, Polysaccharide and Protein Content Before and After Ethanol Precipitation

Since the APTT prolongation activity of the AGP extract was the most significant, APTT was chosen as an index for the next study. In order to further investigate the component properties of the anticoagulant-active substances, the proteolysate of female AGP extract (at 25, 50, and 121 °C) was subjected to ethanol precipitation. The protein and polysaccharide content in the supernatant and precipitate was investigated, as well as their effects on APTT. Figure 5 indicates that the precipitated fraction demonstrated the strongest APTT prolongation activity. This was consistent with the results of ultrafiltration in Figure 2.

Figure 6 shows that the precipitated fraction had the highest polysaccharide and protein content, corresponding to the strongest APTT prolongation activity. This result further proves that the larger MW proteo-polysaccharide was the anticoagulant component.

### 3.6. Effect of Gonad Defatting on Anticoagulant Activity of AGP Extracts

Besides protein and polysaccharide, abalone gonad contains a certain amount of fat. The total fat content in abalone gonads is reported to be 20.78% (dry basis) [4]. Previous lab studies have shown the presence of 20.23% and 15.95% crude fat in the gonads of female and male abalone. The presence of lipids or fat tends to affect the extraction of water-soluble proteins and polysaccharides. Therefore, it is necessary to examine the effect of defatting on the extract activity. In this study, after defatting with acetone, AGP extract was prepared at 50 °C, and its anticoagulant activity was shown in Figure 7. APTT and TT of the female extract were shorted after defatting treatment. On the contrary, APTT and TT of defatted male extracts were significantly prolonged. 

Acetone could alter the structure and stability of protein by destroying the around hydration layer. And the spatial conformation of the protein is changed, which affects its function and solubility. This may account for the altered the anticoagulant activity of the AGP from the defatted male and female gonad.

### 3.7. Fractionation of Female 50 °C AGP Extract

The Chromatogram curve was plotted with elution volume as the horizontal coordinate, OD280 nm, and NaCl concentration as the vertical coordinate, as shown in Figure 8A. The female 50 °C AGP extract was eluted with NaCl gradient solution, and five isolated fractions were obtained, named fractions I, II, III, IV, and V, respectively. Fraction I was the penetration peak. All the fractions were concentrated, dialyzed, and lyophilized to obtain light yellow powder. They had good solubility in water. The recovery rates of the five fractions were 18.6%, 12.1%, 7.8%, 8.6%, and 6.1%, respectively. The polysaccharide content of each component was higher than the protein content. The polysaccharide content of fraction I and IV was slightly higher than others, but still much lower than AGSP (49.5% polysaccharide, 5.4%protein) [29]. The protein content of fraction V was higher than others, and that of AGSP was the lowest (Figure 8B). However, fraction V demonstrated the strongest anticoagulant activity, while the fractions with low protein content and high polysaccharide, such as III, IV, and AGSP, showed the weakest activity. This further suggests that the conjugate protein could not be neglected.

Results of amino acid analysis show (Table 2) that fractions I, II, and III contained 16, 16, and 15 amino acids, respectively, while fractions IV and V were mainly composed of 7 and 4 amino acids. This suggests that Asp, Thr, Ser, and Glu might be involved in the bioactivity of fraction V. DEAE Sepharose Fast Flow is a fast-flow anion exchange column. The last eluted fraction V might be the most negatively charged carrying more ions. Figure 9 shows the infrared (IR) spectra of fractions I, II, III, and V. They seemed similar and showed the characteristic structure of proteoglycans. Only the IR spectrum of fraction IV was different from others. Absorption peak at 789–900 cm^−1^ was characteristic of –d–glucan [32]. The five fractions all have the absorption peak at 1632–1634 cm^−1^ and 1463 cm^−1^, which is the protein characteristic –COO– and –CH_2_–. The absorption peaks at 1036–1038 cm^−1^ and 3250–3500 cm^−1^ in the five fractions are O–H stretching and variable angle vibrations of –OH [33]. Absorption peak at 2927–2940 cm^−1^ is C–H stretching and variable angle vibrations. The IR profiles of the isolated fractions again proved that all five fractions were protein-polysaccharide complexes. Fractions I, II, III, and V showed absorption peaks at 790 and 1297 cm^−1^ due to the presence of the C–S–, suggesting that fractions I, II, III, and V were sulfate polysaccharides. Fraction IV did not have this absorption peak and had the lowest anticoagulant activity, suggesting that this may be due to the lack of sulfate groups as well as the lowest protein content. It was reported that sulfate derivatives of the polysaccharide-protein complex have stronger antitumor activities [34].

### 3.8. Effect of AGP Extract on Activity of Clotting Fator VIII, IX, and XI

APTT has been an effective index used to evaluate the amount and function of clotting factors, such as VIII, IX, XI, and XII, which are involved in the intrinsic coagulation pathway [35]. Clotting factors VIII, IX, and XI are regular items in clinical diagnose of intrinsic disorders of blood coagulation. In this study, the effect of AGP extract on the activity of these three factors was investigated, as shown in Figure 10. Almost all the extract could inhibit the activity of factor VIII and IX, except the sample “50 °C-M-proteolysis 0 h”. As for factor XI, all samples from females presented an inhibitive effect, but all samples from males did not except “50 °C-M”. On the whole, the female AGP extract demonstrated stronger inhibitive activity than the male, this is consistent with the result in Figure 2.

Compared with the “50 °C-F” or “50 °C-M”, their corresponding samples that were exposed to proteolysis showed a significantly weakened inhibitive activity on the three clotting factors. The AGP extract of 50 °C-M even lost the inhibition effect after proteolysis treatment. This result is consistent with the result of Figure 4, proving that protein in AGP is of importance for the bioactivity expression. By contrast, after being heated at 100 °C for 30 min, the inhibitive activity against factor VIII and IX of “50 °C-F” did not change, and the change in “50 °C-M” was not as much as proteolysis. However, for factor XI, proteolysis and heating of both female and male AGP extract could weaken the inhibitive activity of AGP extract. In addition, the ASGP, which contained highest polysaccharide of 49.5% and lowest protein of 5.4%, did not show any inhibitive effect on the three clotting factors. This result further confirmed the positive contribution of the protein on anticoagulant activity.

Interestingly, “121 °C-F” showed the highest inhibition effect on the three clotting factors, followed by “100 °C-F” and “50 °C-F”, which is a reverse order in the APTT prolongation effect, but in the same order of TT prolongation effect, as shown in Figure 2. Therefore, further study is needed to explain this discordance. 

## 4. Discussion

Structure-activity relationship study has been a core issue in polysaccharide research. Previous studies showed that the activity of polysaccharides is affected by the composition of monosaccharide, chain conformation and linkage [36], branched nature, molecular weight size [37], physicochemical properties (such as viscosity and solubility), sulfate content and position [38], etc. In addition, some previous studies have noticed that protein in the proteoglycan complexes also affects the polysaccharide activity. Covalently and non-covalently (electrostatic and other weak interactions, e.g., hydrophobic interactions, van der Waals forces, hydrogen-bonding interactions, etc.) bound protein could influence the structure and function of polysaccharides [39]. Environmental conditions (pH, ionic strength, etc.), in which both are located also influence their interactions.

In protein-bound polysaccharide, the protein links to sugar chain through an O- or N-glycoside bond in the protein-bound polysaccharide [40]. So, it is very difficult to remove the protein portion completely by proteolysis. Proteolysis only occurs inside the protein portion. Although many studies prepared AGP using proteolysis, few concerned the effect of protein on AGP bioactivity. Research on polysaccharides from mushrooms have shown the special role of the protein portion. It is well known that protein–polysaccharide complex, or protein bound polysaccharide, from mushrooms, was used as an immune-stimulating adjunct in cancer clinical chemotherapy, demonstrating beneficial therapeutic effects. For example, the protein-bound polysaccharide from *Coriolus versicolor* COV-1 has been a pharmaceutical for cancer, immune depression, and liver disease [41]. Mizuno et al. obtained 5 hetero-glycan fractions from the fruiting body of *Agaricus hlazei* Murill. One of the fractions, named FIII-2-b (consisting of 50.2% of carbohydrate and 43.3% of protein), exhibited significant antitumor activity. It is of great interest to clarify if the active site of this glycoprotein is located on its polysaccharide portion or on the protein portion. They found that the activity is not from the polysaccharide or the protein portion, but is a result of the complex formed by the polysaccharide portion binding with the protein portion [42]. In addition, similar research also showed that polysaccharide fractions with high protein content exhibited stronger tumor-inhibiting activity [43,44]. Thus, it is clear that the protein portion plays an important role in the bioactivity expression of protein-bound polysaccharide. Zhang et al. and Peng et al. investigated the antitumor activity of water-soluble polysaccharides from *Pleurotus tuber* regium and *Ganoderma tsugae* mycelium, respectively [45,46]. Their studies also supported the positive effect of protein content on the bioactivity improvement of the polysaccharide. A study by Surenjav et al. also revealed that antitumor activities of the proteoglycan from four kinds of fruiting bodies of *LentinusEdodes* were strongly related to the content of bound protein. The single chain β-glucan part of the proteoglycan hardly exhibits antitumor activity. The conjugate protein is favorable to enhancing the bioactivity [47]. It was found that, compared with the chitosan obtained after deproteinization for 5–30 min, the chitosan from deproteinization for 0 min showed similar or greater antibacterial activity against *E. coli*, *B. megaterium*, *B. cereus*, and *S. aureus* [48]. Song et al. also found protein had an effect on the solubility as well as the activity of polysaccharides from lychee nuclear [49].

In this study, AGP extract showed significant APTT and TT prolonging activity, and clotting factor (VIII, IX, and XI) inhibitive effect. Ultrafiltration and ethanol precipitation proved that the active compound lies in the protein-conjugate polysaccharide. However, the activity was weakened a lot, even disappeared, immediately after adding neutral protease into the extract solution. Therefore, it is very clear that the bound protein portion plays a role. To compare with the AGP in this study, a previously reported AGP, AGSP was prepared by multiple proteolysis, ethanol precipitation and anion exchange resin absorption [29]. It was found that AGSP only showed a weak APTT prolongation activity, without inhibiting effect on the above clotting factors.

Moreover, a certain level of sulfate polysaccharide was detected in the AGP extract (data not supply). It is known that the sulfate group in AGP might also be involved in the anticoagulant activity [23]. Related study also reported that ionic interactions of sulfated polysaccharides with proteins are assumed to play a major role for bioactivity expression [50]. Therefore, how to obtain the sulfated and protein-conjugate polysaccharide is a critical point that the industry and academics need to consider. 

Interestingly, the AGP samples, which experienced heating at 100 and 121 °C, demonstrated significant APTT and TT prolongation effect (Figure 2 and Figure 3) and strongest clotting factor inhibition effect (Figure 10). This implies the heat-stability of the abalone protein-conjugate polysaccharide. It has been proved that protein glycosidation could overcome the problem of protein instability when exposed to heat [51,52]. In the protein-polysaccharide complex, the steric hindrance of the polysaccharide chain might inhibit the entanglement of the unfolding protein chain, so that heat denaturation of a single protein chain may become reversible. Thus, the thermal stability of the protein increases, and the linked polysaccharide part acts as the protein “chaperones” in this process. For AGP, the conjugate protein might have a beneficial effect on AGP bioactivity, and the polysaccharide chain increase the stability of the conjugate protein.

## 5. Conclusions

The positive role of the conjugate protein on the anticoagulant effect of AGP was demonstrated. Deproteinization is not advisable for AGP preparation, at least for anticoagulant activity screening and product development. Female gonads could be used for AGP extraction directly, and male gonads need to be defatted by acetone before use. For the female AGP extract, the anticoagulant fraction is mainly located at 30–100 kDa. Ultrafiltration or fractionated precipitation might be preferable to enrich the active portion and discard the potential coagulation accelerator at the part of MW > 100 kDa and <30 kDa. Therefore, anticoagulant AGP extract could be prepared by one-step water extraction and one-step ultrafiltration or fractionated precipitation. Neglecting the deproteinization step, which means much chemical reagent, process time consumption, and voluminous waste water discharge, would obviously reduce the cost of polysaccharide production. Compared with the AGP extracted at 50 and 100 °C, the 121 °C extract demonstrated significant APTT and TT prolongation activity and the highest inhibition effect on the three clotting factors. Therefore, 121 °C AGP extract might be optimal in view of functional product development.

## Figures and Tables

**Figure 1 foods-13-04003-f001:**
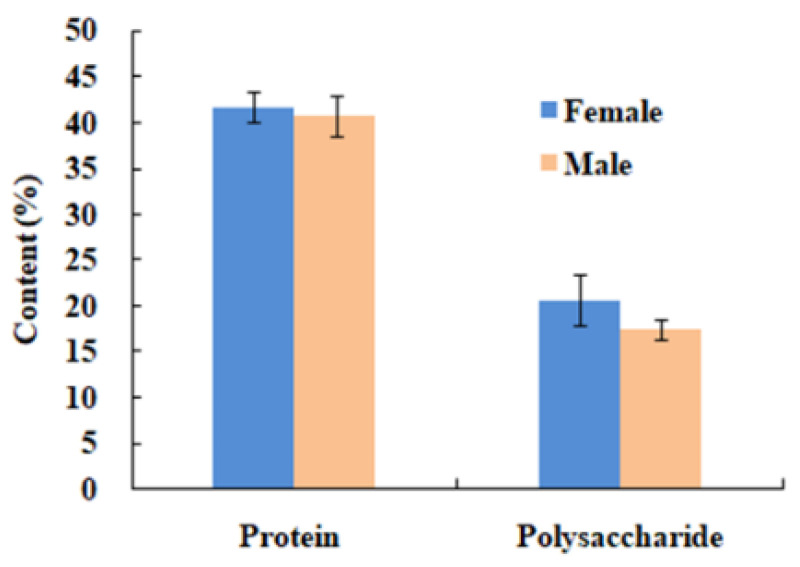
Protein and polysaccharide content (dry basis) in female and male abalone gonad powder.

**Figure 2 foods-13-04003-f002:**
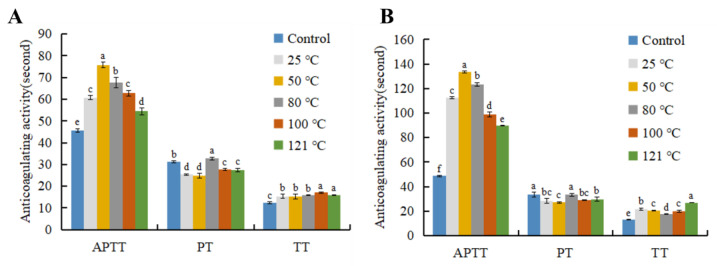
Effects of extraction temperature on anticoagulant activity of abalone gonad polysaccharide extract from female (**A**) and male (**B**). Different lowercase letters stand for significant statistical differences among all the samples in each index.

**Figure 3 foods-13-04003-f003:**
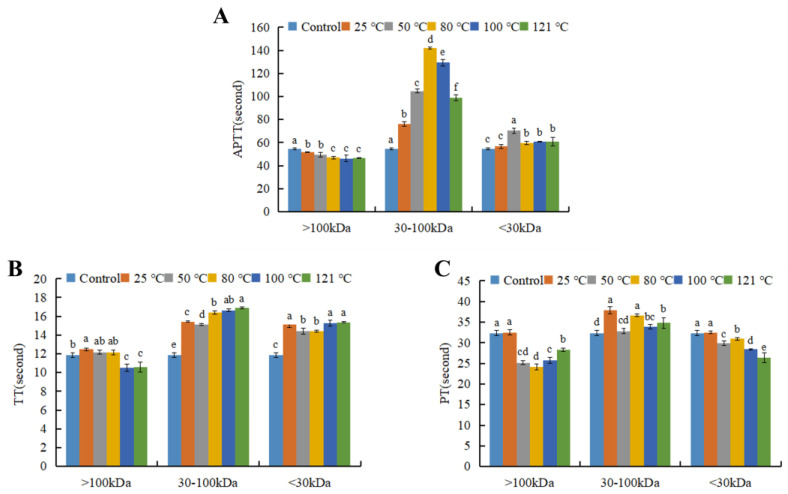
Anticoagulant activity of three fractions obtained from abalone gonad polysaccharide extracts by ultrafiltration. (**A**) APTT, (**B**) TT, (**C**) PT. Different lowercase letters stand for significant statistical differences among all the samples in each molecular range.

**Figure 4 foods-13-04003-f004:**
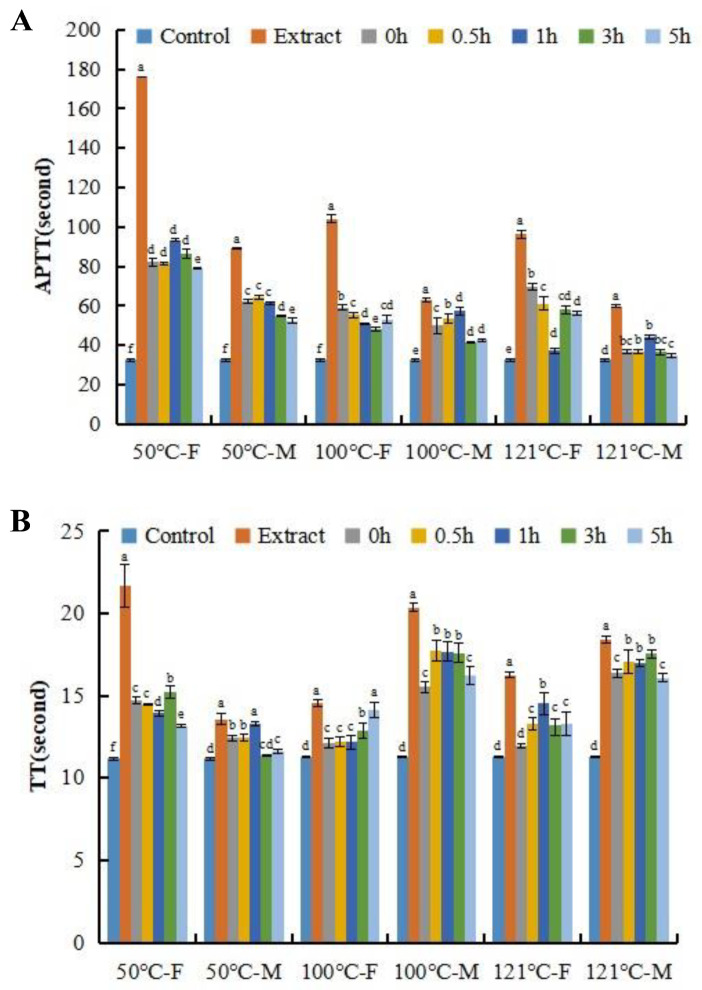
Effects of proteolysis on anticoagulant activity of abalone gonad polysaccharide extracts. (**A**) APTT, (**B**) TT, (**C**) PT. Different lowercase letters stand for significant statistical differences among all the samples in each group, such as the 50 °C-F group. F stands female; M stands male; 0 h, 0.5 h, 1 h, 3 h, 5 h stands proteolysis lasting time. Extract stands for the original extract without proteolysis.

**Figure 5 foods-13-04003-f005:**
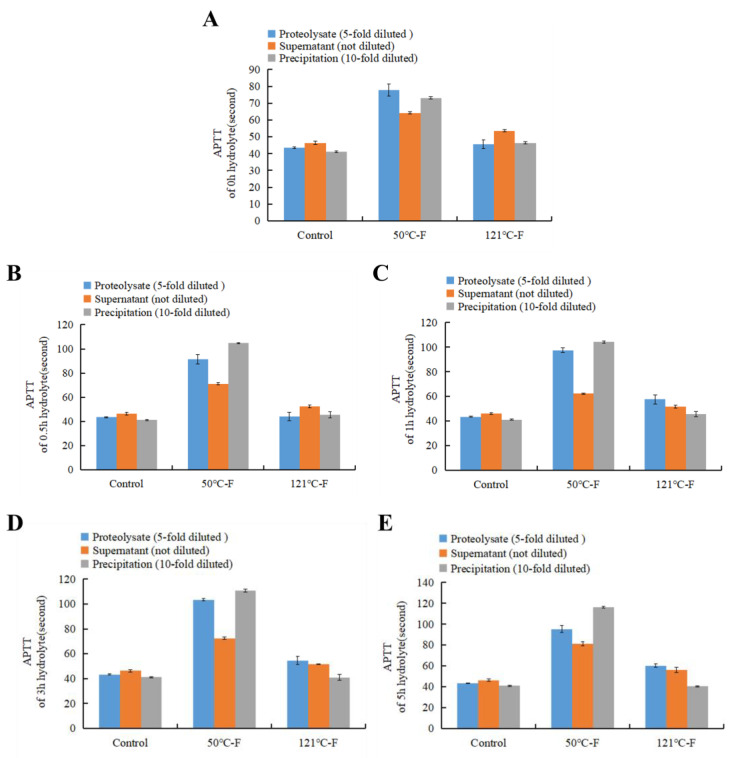
Anticoagulant activity before and after ethanol precipitation of proteolysate after 0 h (**A**), 0.5 h (**B**), 1 h (**C**), 3 h (**D**), and 5 h (**E**); 50 °C-F and 121 °C-F stand for female abalone gonad polysaccharide extract prepared at 50 °C, and 121 °C, respectively.

**Figure 6 foods-13-04003-f006:**
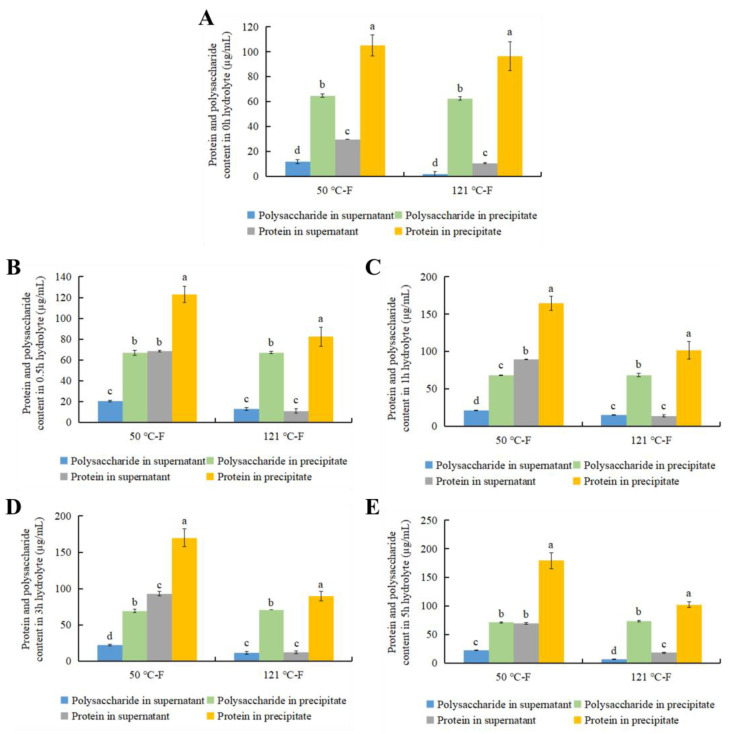
Polysaccharide and protein content in the supernatant and precipitate of the proteolysate, which was obtained after the female abalone gonad polysaccharide extract was proteolyzed for 0 h (**A**), 0.5 h (**B**), 1 h (**C**), 3 h (**D**), 5 h (**E**); 25 °C-F, 50 °C-F, and 121 °C-F stand for female extract prepared at 25 °C, 50 °C, and 121 °C, respectively. Different lowercase letters stand for significant statistical differences among all the samples in each index.

**Figure 7 foods-13-04003-f007:**
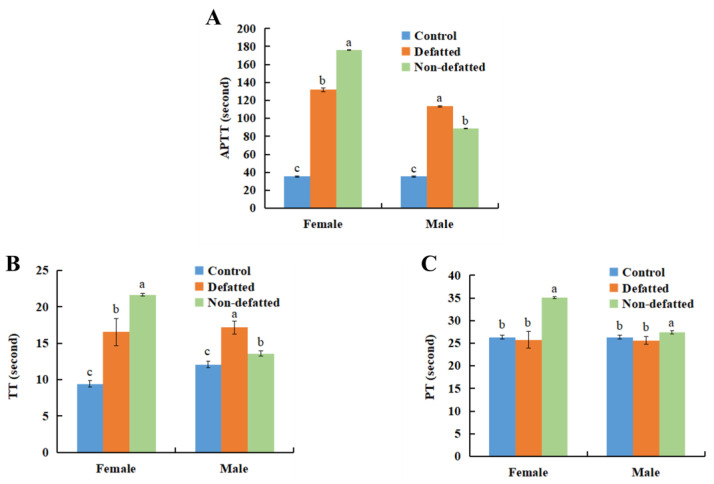
Effects of defatting on anticoagulant activity of abalone gonad polysaccharide extract prepared at 50 °C. (**A**) APTT, (**B**) TT, (**C**) PT. Different lowercase letters stand for significant statistical differences among the three samples of females or males.

**Figure 8 foods-13-04003-f008:**
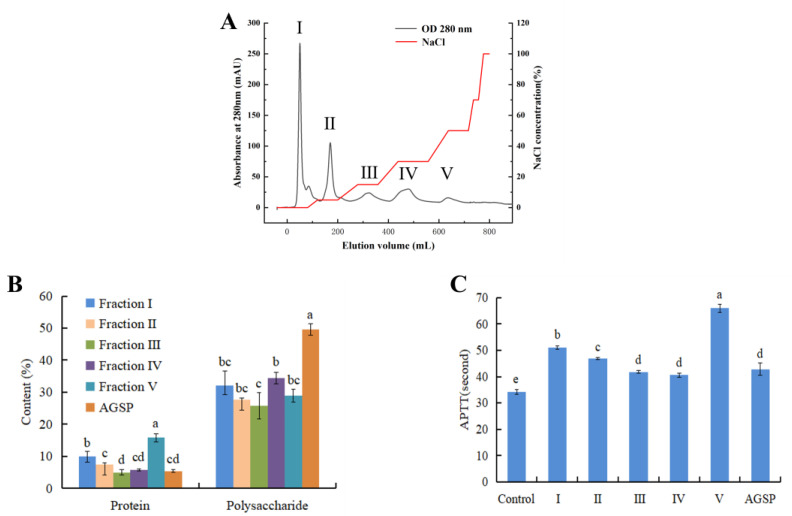
Elution curve of 50 °C female abalone gonad polysaccharide extract on DEAE Sepharose Fast Flow column (**A**), polysaccharide and protein content of each fraction (**B**), and effect of the fractions and AGSP (40 µg/mL) on APTT (**C**). Different lowercase letters stand for significant statistical differences.

**Figure 9 foods-13-04003-f009:**
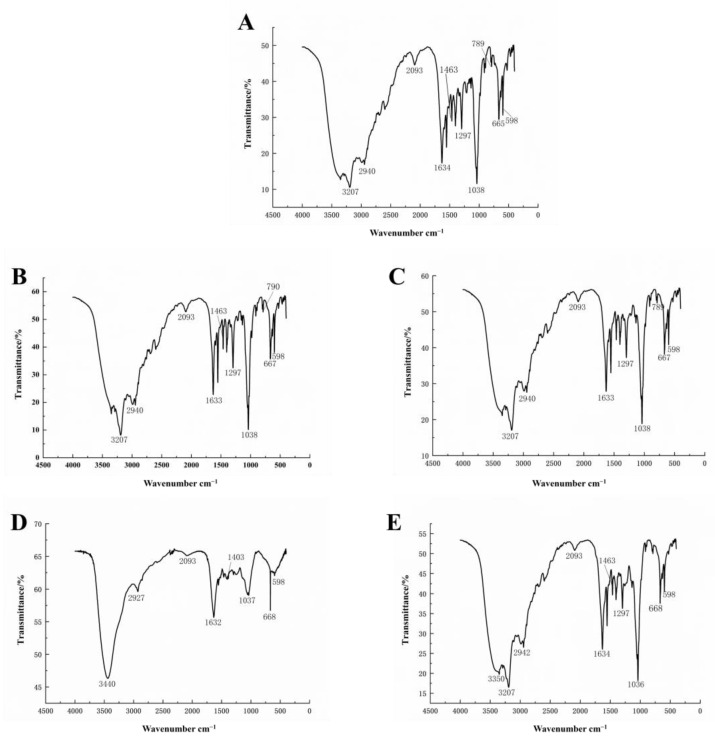
Fourier transform infrared spectra of the five fractions obtained on DEAE Sepharose Fast Flow column. (**A**–**E**) are fractions I, II, III, IV, and V, respectively.

**Figure 10 foods-13-04003-f010:**
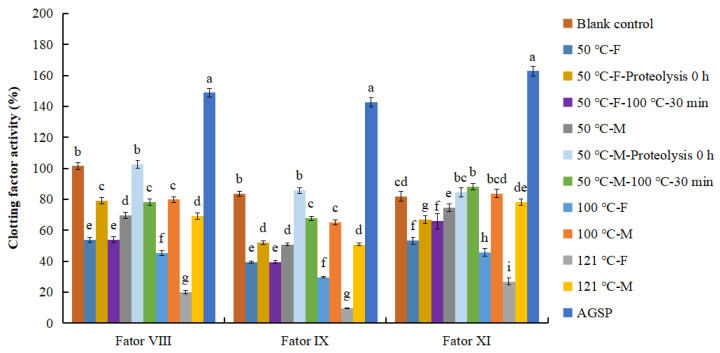
Effects of abalone gonad polysaccharide (AGP) extracts on the activity of clotting factors VIII, IX, and XI; 50 °C, 100 °C, and 121 °C are the temperatures for AGP preparation; F and M mean female and male gonad extract. “50 °C-F-proteolysis 0 h” means the female extract prepared at 50 °C was subjected to proteolysis for 0 h (heating the mixture right after adding neutral protease into the extract); “50 °C-F-100 °C-30 min” means the female extract prepared at 50 °C was subjected to heating for 30 min. Different lowercase letters stand for significant statistical differences among the three samples of females or males.

**Table 1 foods-13-04003-t001:** Amino acid composition of female and male abalone gonad.

Amino Acid	Male (%)	Female (%)
Asp #	9.47 ± 0.13 ^a^	10.10 ± 0.12 ^b^
Thr *	5.60 ± 0.10 ^a^	6.30 ± 0.09 ^b^
Ser	6.29 ± 0.06 ^b^	6.82 ± 0.11 ^a^
Glu #	9.22 ± 0.15 ^a^	10.32 ± 0.08 ^b^
Gly #	9.41 ± 0.2 ^a^	9.26 ± 0.14 ^a^
Ala #	7.86 ± 0.09 ^a^	7.71 ± 0.12 ^a^
Val *	5.33 ± 0.03 ^a^	5.73 ± 0.16 ^b^
Met *	0.95 ± 0.01 ^a^	1.40 ± 0.09 ^b^
IIe *	3.22 ± 0.12 ^b^	4.68 ± 0.10 ^a^
Leu *	5.64 ± 0.07 ^a^	7.05 ± 0.11 ^b^
Tyr #	2.00 ± 0.14 ^a^	2.23 ± 0.14 ^a^
Phe *#	3.03 ± 0.11 ^a^	3.28 ± 0.12 ^b^
Lys *	8.83 ± 0.07 ^a^	6.99 ± 0.08 ^b^
His	2.80 ± 0.12 ^b^	1.99 ± 0.09 ^a^
Arg	6.01 ± 0.09 ^b^	4.45 ± 0.16 ^a^
Pro	0.81 ± 0.02 ^a^	0.99 ± 0.13 ^a^

* essential amino acid; # flavor amino acid; different lowercase letters in the same line stand for significant statistical differences between female and male gonads.

**Table 2 foods-13-04003-t002:** Amino acid composition of each fraction obtained on the DEAE Sepharose Fast Flow column.

Amino Acid	Amino Acid Percentage (%)				
	I	II	III	IV	V
Asp	6.91 ± 0.13	10.59 ± 0.21	6.06 ± 0.09	13.82 ± 0.12	30.23 ± 0.17
Thr	6.10 ± 0.10	6.76 ± 0.12	11.10 ± 0.05	16.00 ± 0.11	22.96 ± 0.12
Ser	3.40 ± 0.11	6.69 ± 0.07	8.66 ± 0.09	14.10 ± 0.07	19.53 ± 0.17
Glu	11.81 ± 0.11	9.87 ± 0.10	10.72 ± 0.12	11.33 ± 0.09	26.94 ± 0.21
Gly	14.97 ± 0.16	11.82 ± 0.12	8.65 ± 0.09	15.55 ± 0.21	0.03 ± 0.04
Ala	9.86 ± 0.12	6.78 ± 0.08	7.27 ± 0.13	14.07 ± 0.17	0.05 ± 0.01
Val	6.40 ± 0.09	4.97 ± 0.12	5.89 ± 0.06	12.43 ± 0.16	0.03 ± 0.01
Met	1.14 ± 0.13	0.67 ± 0.11	-	-	-
IIe	3.79 ± 0.07	2.14 ± 0.07	2.86 ± 0.12	0.01 ± 0.00	0.01 ± 0.00
Leu	5.54 ± 0.09	4.56 ± 0.09	5.63 ± 0.11	0.02 ± 0.01	0.03 ± 0.01
Tyr	0.69 ± 0.12	1.53 ± 0.12	2.34 ± 0.08	-	0.02 ± 0.01
Phe	2.32 ± 0.11	5.36 ± 0.11	5.58 ± 0.09	0.02 ± 0	0.04 ± 0.01
Lys	8.08 ± 0.09	7.62 ± 0.13	5.88 ± 0.13	0.02 ± 0.01	-
His	10.90 ± 0.09	14.68 ± 0.15	12.73 ± 0.12	0.06 ± 0.02	0.09 ± 0.02
Arg	1.81 ± 0.12	2.52 ± 0.16	2.28 ± 0.11	-	0.01 ± 0.00
Pro	5.12 ± 0.14	2.37 ± 0.08	3.14 ± 0.09	0.01 ± 0.00	0.02 ± 0.01

“-” stands not detected.

## Data Availability

The original contributions presented in this study are included in the article; further inquiries can be directed to the corresponding authors.

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
