# Peer review of "Anticoagulant Activity of the Polysaccharide Fromgonad of Abalone Haliotis discus hannai Ino: The Role of Conjugate Protein"

_foods, 2024, doi:10.3390/foods13244003_

Round 1

Reviewer 1 Report

Comments and Suggestions for Authors

The study addresses an interesting topic and presents consistent data. However, some sections are unclear and could be improved. Below are recommendations for corrections and improvements:

  1. The authors mention extractions performed at different temperatures. How were these conducted? Was an oven used? A sealed flask? An autoclave?

  2. The term "neutral protease" should be more specifically described. What is the brand? The type? Clearly specify which protease was used.

  3. In lines 237 to 239, the authors state: "Since the APTT prolongation activity of the proteolysate and the precipitated fraction was so strong, the two parts were diluted 5 and 10 times, respectively, while the supernatant was not diluted." It is unclear whether this is more of a methodological issue than a characterization one. Why not perform the assay in diluted form and express the corrected data for better comparative analysis? As it stands, Figure 5 is equally confusing.

  4. The figure legends should include complete information, avoiding abbreviations or codes. For example, “AGP.”

Author Response

The study addresses an interesting topic and presents consistent data. However, some sections are unclear and could be improved. Below are recommendations for corrections and improvements:

Comments 1: The authors mention extractions performed at different temperatures. How were these conducted? Was an oven used? A sealed flask? An autoclave?

Response 1: Extractions at 25, 50, 80, and 100 ℃ were conducted using sealed flask using water bath. And extraction at 121 was conducted in an autoclave.   

Comments 2:The term "neutral protease" should be more specifically described. What is the brand? The type? Clearly specify which protease was used.

Response 2: The neutral protease used in this study was from Solarbio, with an activity of 50 kU/g. The proteolysis was conducted at 50 ℃ at pH7.0 with the final neutral protease activity of 3000 U/g (protein). We have added this information in the manuscript.

Comments 3:In lines 237 to 239, the authors state: "Since the APTT prolongation activity of the proteolysate and the precipitated fraction was so strong, the two parts were diluted 5 and 10 times, respectively, while the supernatant was not diluted." It is unclear whether this is more of a methodological issue than a characterization one. Why not perform the assay in diluted form and express the corrected data for better comparative analysis? As it stands, Figure 5 is equally confusing.

Response 3: Thank you for this advice.

This sentence has been moved to the method part. Assays in this part were performed in diluted form.

In this study, the APTT prolongation activity (not the polysaccharide yield) of the extract is the most important index we follow to discuss the relationship between the activity and conjugated protein. The proteolysate (the mixture after proteolysis) and the precipitated fraction were diluted 5 and 10 times, respectively, and a moderate APTT value could be obtained.

For polysaccharide and protein quantification in the extract, data obtained from the diluted sample could be multiplied by dilution folds (in Figure 6). But the APTT prolongation activity is not reasonable to multiply. So we have to present the data of APTT prolongation activity of the diluted sample directly in Figure 5. Although different dilution fold seems not convenient for comparative analysis, but it is very clearly that most of the active part is located in the precipitated fraction, which contained macromolecules, protein-conjugated polysaccharide.  

Comments 4:The figure legends should include complete information, avoiding abbreviations or codes. For example, “AGP.”

Response 4: Thank you for this advice. Complete form of AGP was used in the figure legends which concerned AGP.

Reviewer 2 Report

Comments and Suggestions for Authors

This manuscript reported the anticoagulant activity of  polysaccharide isolated from gonad of abalone Haliotis Discus Hannai Ino. The authors applied many treatment methods such as defatting, effects of temperature , proteolysis, ultrafiltration and, ethanol precipitation to examine the role of  the conjugated protein on anticoagulant activity. Male and female animal samples were also used to see the efficiency. Overall, it is an important study to understand the anticoagulant activity of abalones. The observed results will be useful to understand the biological activities of the polysaccharides isolated from abalones. The introduction, materials and methods, results and discussion section are prepared well with enough details. Though the manuscript is technically sound, it needs further improvement. I have appended the following points for the attention of the authors

1. Manuscript needs careful editing in introduction and methodology section. Text in section 2.3 of methodology part needs attention for editing

2. Species name in the title and also in the text should be changed to small letters (Haliotis discus hannai Ino). 

3. Please provide high resolution images for figures 5, 6 and 9. The font size of axis title or legends are very low, hence difficult to understand the values. 

4. The membrane size cut off used for filtration looks too large for polysaccharides. Please provide references for this size or give a justification for using these three MW cut off membranes.  

5. Provide reference for Mizunoet al. in the text (see page 13, line 369). 

Author Response

Comments 1: Manuscript needs careful editing in introduction and methodology section. Text in section 2.3 of methodology part needs attention for editing

Response 1: We have edited the introduction and methodology sections, deleting some part not very necessary and adding some information. Section 2.3 of methodology part was also re-edited.

Comments 2: Species name in the title and also in the text should be changed to small letters (Haliotis discus hannai Ino). 

Response 2:Thank you for this advice. Abalone species name in the title and the text have been changed to small letters (Haliotis discus hannai Ino)

Comments 3: Please provide high resolution images for figures 5, 6 and 9. The font size of axis title or legends are very low, hence difficult to understand the values. 

Response 3: Figure 5,6,9 have been replaced by higher resolution images. And the font size of the axis title is small, we will enlarge them accordingly.

Comments 4:The membrane size cut off used for filtration looks too large for polysaccharides. Please provide references for this size or give a justification for using these three MW cut off membranes.  

Response 4: We provide three references as below.

In this reference (Lin, Z. C.; Pan, X. M.; Wu, Q. C.; Xue, Y.; Huang, J.F.; Pan, Y.T.. Isolation, Purification, Structure Characterization and Antioxidant Activity of Alkali-extracted Polysaccharide from Abalone Viscera[J].Science and Technology of Food Industry,2024,45, 53-60.), alkali-extracted polysaccharides from Abalone viscera (Aavp) were isolated and purified. Four fractions (Aavp Ia, Aavp Ib, Aavp IIa, and Aavp IIb) were obtained from the crude polysaccharide. Because of the highest yield, Aavp IIa was selected for further structural analysis. Aavp IIa was composed of xylose and galactose, with a relative MW of 166513 Da.

  In this reference (Zhu, B.W.; Li, D.M.; Zhou, D.Y.; Han, S.; Yang, J.F.; Li, T.; Ye, W.X.; Greeley, G.H.; Structural analysis and cck-releasing activity of a sulphated polysaccharide from abalone ( Haliotis discus hannai ino) viscera. Food Chem. 2010, 125, 1273-1278.), a fraction of water-soluble sulphated polysaccharide conjugate, termed AHP-2, was obtained from abalone ( Haliotis discus hannai Ino) viscera by protease-assisted aqueous extraction followed by precipitation with ethanol and purification with gel filtration chromatography.  AHP-2 is homogenous with an average MW of about 11.0kDa.

In this reference (Song, S.; Zhang, B.; Wu, SF.; Huang, L.; Ai, CQ.; Pan, JF.; Su, YC.; Wang, ZF.; Wen, CR. Structural characterization and osteogenic bioactivity of a sulfated polysaccharide from pacific abalone (Haliotis discus hannai Ino). Carbohydrate Polymers. 2018, 182, 207-214. ), the MW of the purified sulfated polysaccharide was only 6.6 kDa.

In our study, we didn’t use protease to lower the molecular weight of the protein-conjugated polysaccharide from abalone gonad. So, we tried MW cutoff membrane of 30 and 100 kDa, and obtained three frations, with MW range of <30 kDa, 30-100 kDa and >100 kDa.

Comments 5: Provide reference for Mizuno et al. in the text (see page 13, line 369). 

Response 5: This reference was provided.

Mizuno, T.; Inagaki, R.; Kanao, T.; Hagiwara, T.; Nakamura, T.; Ito, H.; Shimura, K.; Sumiya, T.; Asakura, A. Antitumor activity and some properties of water-insoluble hetero-glycans from Himematsutake, the fruiting body of Agaricus blazei murill [J]. Agric. biol. chem. 1990, 54, 2897-2905.